# Pyridine Carboxamides Based on Sulfobetaines: Design, Reactivity, and Biological Activity

**DOI:** 10.3390/molecules27217542

**Published:** 2022-11-03

**Authors:** Eugene P. Kramarova, Sophia S. Borisevich, Edward M. Khamitov, Alexander A. Korlyukov, Pavel V. Dorovatovskii, Anastasia D. Shagina, Konstantin S. Mineev, Dmitri V. Tarasenko, Roman A. Novikov, Alexey A. Lagunin, Ivan Boldyrev, Aiarpi A. Ezdoglian, Natalia Yu. Karpechenko, Tatiana A. Shmigol, Yuri I. Baukov, Vadim V. Negrebetsky

**Affiliations:** 1Department of Medical Chemistry and Toxicology, Pirogov National Research Medical University, Ministry of Health of Russia, 117997 Moscow, Russia; 2Laboratory of Physical Chemistry, Ufa Institute of Chemistry, Ufa Federal Research Center, Russian Academy of Sciences, 450071 Ufa, Russia; 3National Research Center “Kurchatov Institute”, 123098 Moscow, Russia; 4Shemyakin-Ovchinnikov Institute of Bioorganic Chemistry, Russian Academy of Sciences, 117997 Moscow, Russia; 5N.D. Zelinsky Institute of Organic Chemistry, Russian Academy of Sciences, 119991 Moscow, Russia

**Keywords:** derivatives of pyridine carboxylic acids, sulfobetaines, mechanism of reaction, NMR and FT-IR spectroscopy, X-ray, quantum-chemical calculations

## Abstract

The synthesis of the products of the 1,3-propanesultone ring opening during its interaction with amides of pyridinecarboxylic acids has been carried out. The dependence of the yield of the reaction products on the position (*ortho*-, *meta*-, *para*-) of the substituent in the heteroaromatic fragment and temperature condition was revealed. In contrast to the *meta*- and *para*-substituted substrates, the reaction involving *ortho*-derivatives at the boiling point of methanol unexpectedly led to the formation of a salt. On the basis of spectroscopic, X-Ray, and quantum-chemical calculation data, a model of the transition-state, as well as a mechanism for this alkylation reaction of pyridine carboxamides with sultone were proposed in order to explain the higher yields obtained with the nicotinamide and its *N*-methyl analog compared to *ortho* or *meta* parents. Based on the analysis of ESP maps, the positions of the binding sites of reagents with a potential complexing agent in space were determined. The in silico evaluation of possible biological activity showed that the synthetized compounds revealed some promising pharmacological effects and low acute toxicity.

## 1. Introduction

For many years, our group has focused on discovering and developing new heterocyclic compounds that have nootropic, anti-inflammatory, and antidepressant activities. Our main goal is to develop effective medications for treating cognitive impairments, including ischemic stroke and its complications [1,2,3,4]. The recent COVID-19 pandemic highlighted the emerging need for these kinds of therapeutic agents. Moreover, the greatest danger to the patient’s mental health is not the disease itself, but the consequences it causes. Currently available medications have not been able to deal with emerging post-COVID-19 disorders alone. One of the reasons is insufficient bioavailability and multiple side effects. Previously, we managed to increase the bioavailability of Phenotropil by introducing chemical modification, which led to the production of a water-soluble form based on taurine [1]. The same approach was shown for homotaurine [3].

In continuation, in this work we studied the synthetic possibilities of the propansultone ring opening reaction, leading to the formation of sulfobetaines as end products.

Sulfobetaines are key structural features of redox systems [5], polymers, surfactants [6], medicine [7,8,9,10,11,12], nanoparticles, and immobilization adjuvants [13]. Sultones were used to construct cellulose membranes with superior antifouling properties and biocompatibilities [14], tailor-made zwitterionic lignin for resistance to protein adsorption [15], amphiphilic copolymers containing zwitterionic sulfobetaine as pH and redox responsive drug carriers [16], and inhibitors of squalene synthase [17].

To support the various fields of application of sulfobetaines, numerous methods have been developed for their preparation using sultone from various substituted pyridines, such as pyridine carboxylic acids, bipyridines, and others [13,18,19,20,21,22,23,24,25,26]. For the synthesis of the pyridinecarboxylic acids studied in this work, containing a homotaurin (tail), we used the previously proposed method [27].

The reaction comes from the opening of the sultone ring, as shown in Figure 1.

In contrast to the sultone ring formation (which could be considered as well described [24]), the chemistry of the sultone ring opening remains unclear. In the current study, we aim to study the subtle features of the reaction presented in Figure 1, including the influence of the structure of the substrate and temperature on its mechanism. We further aim to evaluate the possible anti-inflammatory activity of the synthesized compounds.

The aim of this study was to study and characterize the compounds presented in Table 1.

## 2. Results and Discussion

The reaction yields of sultone alkylation of pyridine carboxamides are presented below (Table 1).

Yield analysis revealed that, when an amide group (substrates **3**–**7**) is in a *meta*- or *para*-position, it increases reaction yield, while in *ortho* positions (substrates **1**, **2**) reaction yield decreases.

To further investigate this phenomenon, we combined spectroscopic, X-Ray, and quantum-chemical calculation data.

### 2.1. Synthetic Aspects and Reactivity

#### 2.1.1. *Ortho*-Pyridine Carboxamides

We have collected IR-spectra of the starting compounds (Figure 1).

There is a notable difference in the position of the amide-II peak (NH bending) of *ortho*- and *para*-/*meta*-derivatives. The first is being right-shifted. Moreover, the shift of the amide-II peak of compound **2** is bigger. The shift of the NH bending peak is due to the formation of the hydrogen bond. The last is stronger in the case of compound **2** than in the case of compound **1**.

To understand the structure and the dynamic behavior, we investigated compounds **1** and **2** using 2D NMR approaches. Implementation of HSQMBC techniques allowed measurement of the vicinal ^3^*J*_HC_ couplings. Namely, in compound **1** the amide proton at 7.61 ppm is in the trans-position to the carbon of the pyridine ring (149.4 ppm), with ^3^*J*_HC_ equal to 8 Hz, while the ^3^*J*_HC_ of the second amide proton at 8.10 ppm is too small to give rise to a detectable cross-peak in HSQMBC. The amide proton in compound **2** (8.07 ppm) is in cis-position to the carbon of the aromatic ring, because no cross-peak is observed in HSQMBC, while the ^2^*J*-coupling to the carbonyl (~7 Hz) is measurable (Figure 2).

The NH_2_-group protons of compound **1** experience the slow flips—their signals become broader upon heating and exchange cross-peaks are observed in both NOESY and ROESY spectra, which follows from the identical sign of cross- and diagonal peaks. We assessed the rate of such flips using the equations of ZZ-exchange spectroscopy and found that k = 0.21 ± 0.01 s^−1^.

For the case of compound **2**, only the single NH and N-methyl group signals are observed, which implies that amide bond flips either do not take place or the population of the cis-conformation is lower than the detection limit.

Estimated as a result of quantum-chemical calculations using the M052X-D3/TZVP (methanol PCM, implicit) method, the N–H hydrogen bond breaking energy was 15.3 kJ mol^−1^.

We conclude that steric hindrance of the pyridine nitrogen atom with the amide hydrogen atom of the carbamoyl group is the second point (along with low ESP) that decreases the yield of the reaction.

#### 2.1.2. *Meta*- and *Para*-Pyridine Carboxamides

All *para*-derivatives (compounds **6**, **7**) and meta-diethylamide pyridine (compound **5**) show moderate yields in sultone alkylation (products **13**, **14**), while *meta*-amide (compound **3**) and *meta*-methylamide (compound **4**) demonstrate sufficiently higher yields (products **10**, **11**) (Table 1). Because *meta*- and *para*-derivatives in contrast to *ortho*-derivatives have no geometrical constraints, it could be suggested that there is some additional activation during the path of the reaction. To find out what it is, we have performed continuous monitoring of the reaction using IR spectroscopy. It appears that, in the most high-yield reaction (namely reaction of compounds **3**, **4**, Table 1), the carbonyl group is involved. The C=O stretching band of compound **4** during the reaction shifts from 1640 to 1670 cm^−1^ (Figure 3).

In the case of compound **3**, along with the shifting of C=O stretching from 1675 cm^−1^ to 1690 cm^−1^, an additional band of 1645 cm^−1^ arises (Figure 3). The last may point to two different stable conformers of the product. In the case of compound **5**, the product formation does not affect the C=O stretching band (Figure 3). The reason for this is the absence of the proton in the amide group in this derivative.

The hydrogen bonding of the amide group could be of intermolecular or intramolecular origin. From X-Ray data (see above) it follows that, in the crystal state, the sulfo group forms an intermolecular hydrogen bond with the amide group. To check if the observable band shift is caused by the internal or external hydrogen bonding, we have collected IR spectra of the two products in different dilutions (Figure 4).

One idea is that, upon dilution, intermolecular hydrogen bonds should disappear while intramolecular hydrogen bonds should remain intact. It appears that the IR spectra of compounds **10** and **11** remain intact upon dilution. Amid bands do not shift. Thus, we conclude that hydrogen bonds between the sulfo and amide groups are internal (intramolecular) and propose the following mechanism of reaction activation in the case of *meta*-pyridine carboxamides with the NH group (Figure 2).

The reaction starts from the attack of sultone on the N atom. Once the intermediate is formed, the hydrogen bond appears between the sultone in-ring oxygen atom and an amide hydrogen atom. The hydrogen bond is assumed to either decrease the energy barrier of the sultone ring opening or to stabilize the transition state. This is the reason for the increased reaction yield in the case of *meta*-pyridine carboxamides.

#### 2.1.3. Investigations under High Temperature

The study of the effect of temperature on the yield of the target product (Figure 1) was carried out in boiling methanol for 4 h [28,29]. In the case of substrates with meta- and para-orientation of the amide fragment (compounds **3**, **6**), the yield of the final product (compounds **10**, **13**) practically did not change (see Experimental part).

At the same time, for substrates with ortho-orientation of the amide fragment (compounds **1**, **2**), a similar reaction unexpectedly led to the formation of salt (compounds **15**, **16**) (Figure 3).

Compound **15** was also synthesized in methanol-*d*_4_. The signal (s) at 3.23 ppm, which corresponds to the **CH_3_O**-CH_2_ group in the CH_3_O(CH_2_)_3_SO_3_^−^ anion, disappears in the ^1^H NMR spectrum. The degree of deuteration of the OCD_3_ group is > 98%. As a result of the exchange processes with the solvent, mobile acidic protons SO_3_H(D) and C(O)NH_2_(D_2_) are partially replaced by deuterium (deuteration degree ~50%) (see Appendix A).

The data of quantum chemical calculations (M052X-D3/TZVP) confirm the proposed scheme. Indeed, for compound **1**, the conformation in which the oxygen atoms of the amide group and the nitrogen atoms in the pyridine ring are located in the same plane turned out to be preferable due to stabilization by an intramolecular hydrogen bond (Figure 3). When heated, the breaking of the hydrogen bond (structure **A**) is accompanied by protonation with the formation of a cation (structure **B**), which is characterized by a conformation in which the oxygen and nitrogen atoms are located opposite each other. For compounds **3** and **6**, the position of the substituent affects the energy characteristics of the cations to a lesser extent (see Appendix A). In addition, attention is drawn to the increase in the values of charges on the nitrogen atom in the pyridine ring in the series of compounds **1**, **3**, and **6** in the following sequence: −0.21; −0.15; −0.14. In other words, the nitrogen atom in the pyridine ring of *ortho*-derivatives more readily accepts the H^+^ cation compared to compounds **3** and **6**.

### 2.2. X-ray Investigation

The structures of compounds **9–14** were studied by single crystal X-ray diffraction (Figure 5A).

The values of bond length and angles of the abovementioned compounds are very close to organic compounds with similar fragments [30]. The substituted pyridine ring and amide CONR_1_R_2_ moieties in molecules **10, 11**, and **13** are coplanar. In the rest of the compounds, carbonyl atoms are deviated from the plane of the pyridine ring (the angle of these moieties are equal to 45.7, 68.3, and 17.8° for **9**, **12**, and **14**, respectively). The presence of amino groups leads to the formation of 1D H-bonded motifs in crystal packing (Figure 5B). At the same time, the crystal packing of **11** molecules form centrosymmetric dimers via N-H^…^O bonds.

### 2.3. Quantum Chemistry

#### 2.3.1. Electrostatic Potential Analysis

The ring opening of 1,3-propanesultone during its reaction with amides could proceed under charge control [31,32]. Analysis of electrostatic potential (ESP) maps makes it possible to find the position of the binding sites of reagents with a potential complexing agent in space. Below are the ESP maps of compounds **1–7** and sultone in the gas phase (Figure 6).

The blue spheres show the ESP extrema corresponding to the regions with electron donor properties. Naturally, the largest extremum corresponds to the region located near the oxygen atom of the carbonyl group. However, the most interesting is the region near the nitrogen atom in the pyridine ring of the substrates. Changing the position of the ring substituent from the *para*- to the *meta*- and *ortho*-positions leads to a reduction in potential dropping from −60.1 kJ∙mol^−1^ (compound **1**, Figure 6) to −138.7 kJ∙mol^−1^ for the *meta*-position (compound **3**) and −147.0 kJ∙mol^−1^ for *para*-substitution (compound **6**). The introduction of a methyl group into the amide fragment of compounds **1**, **3**, and **6** is accompanied by a decrease in the ESP values at the nitrogen atom (compounds **2**, **4**, **7**, Figure 6). The presence of two ethyl fragments in the amide substituent in the *meta*-position (compound **5**) leads to a decrease in the ESP value to −176.8 kJ∙mol^−1^. ESP extrema corresponding to regions exhibiting pronounced electron-withdrawing properties are marked with red spheres in sultone and pyridine substrates **1**–**7**. The regions located near the carbon atom of the sultone are capable of the nucleophilic ring opening of 1,3-propanesultone by substrates **1**–**7**. The extremum regions on the ESP maps of the reagent and substrates can be considered as reaction centers, and the carbon atom located next to the sulfur atom in 1,3-propanesultone can be considered as an electron acceptor in this reaction (Figure 6). Compound 1 has some additional peculiarities, in contrast to other substrates; instead of one local minimum at the reaction center (nitrogen atom), two symmetric points are observed equidistant from the plane of the ring. This may indicate the presence of steric hindrances in nucleophilic addition reactions. A change in the position of substituents in the aromatic fragment of the substrates **1**–**7** affects the electron-donating properties of the reaction center. Indeed, a satisfactory correlation is observed between the charge on the nitrogen atom and the reaction yields. Thus, compounds **3**–**5** (*meta*-substitution), as well as **6**, **7** (*para*-substitution) are characterized by the highest charges and yields of reaction.

#### 2.3.2. Transition States Search

The search for transition states was carried out in two steps: relaxed scanning along the reaction coordinate to find the maximum point (transition state) on the potential energy surface (PES) projection and the subsequent standard procedure for localizing the stationary maximum point. As a result, the transition states of reactions of nucleophilic ring opening of 1,3-propanesultone in the gas phase and in the presence of a solvent were obtained. Figure 7 shows the structure of the transition state of the considered reaction with the participation of substrate **1**, with the formation of compound **8**, the geometric parameters of which are optimized in methanol.

The structures of transition states with the participation of other substrates (Table 1) have a similar structure (see Appendix A).

As can be seen from the example of the transition state in reaction for substrates **1**–**8** (Figure 7), the 1,3-propanesultone ring opens at the C–O bond, followed by coordination through the carbon atom to the nitrogen atom of the pyridine fragment at a distance of slightly more than 2 Å. Note that the reaction center of the reagent (1,3-propanesultone), based on the analysis of the ESP map, has pronounced electron-withdrawing properties (Figure 6).

#### 2.3.3. Thermodynamic and Activation Parameters

Optimization of geometric parameters of reagents, products, and transition states of nucleophilic ring-opening reactions of 1,3-propanesultone with amides of pyridine derivatives made it possible to calculate the total energies of the systems under study and evaluate the thermodynamic and activation parameters of the reactions (Table 2).

As can be seen from Table 2, in all cases the reaction under study has a pronounced exothermic character. However, the Gibb’s energy of reactions **1** → **8** and **2** → **9** is, on average, four times higher than for other reactions. The highest activation barrier is also characteristic of 1,3-propanesultone ring opening reactions with substrates **1** and **2**. Note that the nitrogen atom in the pyridine ring of compounds **1** and **2** is characterized by the lowest electron-donor properties compared to compounds **3**–**7**, according to ESP maps (Figure 6). These results are consistent with the lowest yields obtained with compounds **8** and **9** (Table 1).

For the rest of the reactions, the values of the thermodynamic and activation parameters are generally comparable. The lower values of the Gibb’s free energy of reactions and low activation barriers for the formation of products **10**–**14** in comparison with reactions **1** → **8** and **2** → **9** allow us to conclude that the rate of formation of more stable products of the reaction under consideration is higher. A lower yield of the product reaction **5** → **12** in comparison with the expected one (Table 1 and Table 2) can be caused by the influence of steric factors. The results obtained by MD simulation that do not contradict this assumption are presented in Appendix A.

Interesting patterns were found when comparing the dependence of the product yield on the thermodynamic parameters (Figure 8).

It appeared that the reaction of sultone for *meta*-derivatives differs from the reaction with *para*- and *ortho*-derivatives. While increasing ESP increases reaction yield in the case of *para*- and *ortho*-derivatives, it decreases yield in the case of *meta*-derivatives. At the same time, Gibb’s energy could not be used to assign high reaction yields to the *meta*-derivatives, because *para*-derivatives have lower Gibb’s energy but their reaction yields are also lower.

Thus, the minimum reaction yield obtained for *ortho*-derivatives could be assigned to the lowest ESP on the nitrogen atom of the starting compound. *Meta*-derivatives have increased reaction yield, and the last is hard to assign to higher ESP or decreased Gibb’s energy. To evaluate the question further, we have performed spectroscopic investigations.

### 2.4. Biological Activity

#### 2.4.1. In Silico Evaluation of Possible Biological Activity

The study of possible pharmacological effects was made by the PASS Online web application [33,34,35]. The PASS (Prediction of Activity Spectra for Substances) score is based on structure-activity analysis of a training set that includes structures of tested compounds and data on more than 4000 known therapeutic effects and molecular mechanisms of action. Biological activity is presented in a qualitative way (active/inactive). The chemical structure is described in the form of MNA descriptors (Multilevel Neighborhoods of Atoms—multiple atomic neighborhoods) [34]. The algorithm for creating classification “structure-activity” relationship models is based on the structures of substances of the training set and Bayesian-like estimates [33,34,35]. The average prediction accuracy calculated by a leave-one-out cross-validation procedure for all compounds in the training set and all types of biological activity presented is approximately 95%. The analysis of prediction results revealed the most frequent and probable pharmacological activities for compounds **8**–**14**. They are shown in Table 3.

Table 3 shows that the cerebral anti ischemic effect was predicted with high probability for all compounds. Antianginal activity was also predicted for all compounds with medium likelihood. The effects related to phobic disorder treatment were predicted with high probability for compound **14** and medium probability for compounds **8**, **10**, **11**, and **13**. Several other pharmacotherapeutic effects, including some types of antineoplastic activity, were predicted for some compounds with medium probability.

CLC-Pred web service (Cell Line Cytotoxicity Predictor, http://way2drug.com/Cell-line, accessed on 25 September 2022) was used to evaluate the possible cytotoxicity of compounds in tumor and normal cell lines [36]. CLC-Pred is based on the technology and methods used in PASS Online and predicts cytotoxicity against 278 tumor and 27 normal cell lines, with an average accuracy of 95% calculated by a leave-one-out cross-validation procedure. The training set for CLC-Pred was created based on data from the ChEMBL database. The results of prediction showed that cytotoxicity for normal cell lines was not predicted for the synthesized compounds. The predicted safety indicates the further possibility to use these compounds in drug development. For some cell lines, the toxicity was predicted as moderate (Table 4).

To determine the potential acute toxicity of the synthesized compounds, the LD_50_ values for rats with different routes of administration were predicted using the GUSAR Online acute toxicity web service [37]. It uses four QSAR models for prediction, which correspond to the route of administration of the substance (oral, intravenous, intraperitoneal, and subcutaneous). The models were based on the data from the RTECS database, and their accuracy is sufficient to be used in these types of assessments. Based on LD_50_ values (mg/kg), a toxicity class from 1 to 5 is determined, where class 1 is a highly toxic compound and class 5 is non-toxic. Table 4 shows that all compounds were predicted as non-toxic (class 5) or slightly toxic (class 4).

#### 2.4.2. In Vitro Studies

Sulfobetaines are used as antifouling and antimicrobial materials, as a drug delivery system, and much more [38,39,40,41,42]. Synthesized compounds **8**–**13** were tested for the ability to influence the viability of four cell lines using the MTT assay (Table 5).

Based on MTT assay, all tested compounds have low cytotoxicity against the tested cell lines. We further tested the ability of these substances to exhibit anti-inflammatory activity. The study was carried out on the RAW 264.7 macrophage cell line. Macrophages play a critical role in the development, progression, and resolution of many inflammatory responses. By secreting various cytokines, these cells can direct the immune response towards a more or less pro-inflammatory profile. In our experiment, we treated the RAW 264.7 cell line with lipopolysaccharide (LPS) and measured the levels of expression of interleukin 1β (IL-1β), inducible nitric oxide synthase (iNOS), and cyclooxygenase-2 (COX2) genes involved in inflammation. COX2 and iNOS were chosen as they are the main inflammatory mediators. IL-1β is one of the main pro-inflammatory cytokines. Figure 9 shows real-time PCR data on gene expression for compound **12** (Gene expression results for all compounds are presented in the Appendix A Inflammatory Gene Expression).

According to our results of the LPS-induced RAW 264.7 cell inflammation model, the tested compounds had multidirectional effects (see Appendix A). However, compound **12** had a pronounced anti-inflammatory potential, so with respect to the COX2 gene, iNOS and IL-1b, it caused a dose-dependent inhibitory effect.

## 3. Materials and Methods

### 3.1. General

The purities of compounds were assessed by NMR to be ≥95%. ^1^H, ^2^H, ^13^C, and ^15^N NMR spectra were recorded on 300 MHz (300.1, 46.1, 75.5, and 30.4 MHz, respectively) and 400 MHz (400.1, 61.4, 100.6 and 40.6 MHz, respectively) spectrometers in CDCl_3_, DMSO-*d*_6_, D_2_O, and methanol-*d*_4_ solutions using 0.05% Me_4_Si as the external or internal standard, and CH_3_NO_2_ as the external (or internal for calibration) standard for ^15^N spectra.

Determinations of the structures of the obtained compounds and assignments of ^1^H, ^2^H, ^13^C, and ^15^N signals were made with the aid of 2D COSY, NOESY, DOSY-LED, edited-HSQC, HMBC, ^1^H–^15^N HSQC, and ^1^H–^15^N HMBC spectra. Spin multiplicities were described as *s* (singlet), *d* (doublet), *t* (triplet), or *q* (quartet).

IR spectra in the solid state were recorded on a Bruker Tensor-27 (Bruker Corporation, Bremen, Germany) spectrometer with the attenuated total internal reflectance (ATR) module. Refraction parameters were measured with an IRF-454B2M refractometer (KOMZ, Kazan, Russia). Melting points were determined on a Stuart SMP10 instrument (Barloworld Scientific Ltd., Stone, UK). Elemental analyses were carried out on a ELEMENTAR vario MICRO cube Microanalyzer (Abacus Analytical Systems GmbH, Babenhausen, Germany). All initial reagents and solvents were purchased from Sigma-Aldrich (Saint Louis, MO, USA).

High-resolution mass spectra were obtained on a Bruker maXis Q-TOF instrument (Bruker Daltonik GmbH, Bremen, Germany) equipped with an electrospray ionization (ESI) ion source. The spectra were processed using Bruker Data Analysis 4.0 software.

### 3.2. Dynamic NMR

Samples **1** and **2** were dissolved in CDCl_3_ and placed into the 5 mm NMR tube. To narrow down the signals of amide protons, the temperature was set at 0 °C for **1** and 10 ^°^C for **2**. The structure of the compounds was studied by heteronuclear *J*-couplings and NOESY. *J*-couplings were measured using the pure in-phase HSQMBC pulse sequences, optimized for the 10 Hz couplings [43]. Two spectra–in-phase (IP) and anti-phase (AP) were recorded in the interleaved manner, and the *J*-coupling was measured as the peak distance in the sum (IP+AP) and difference (IP-AP) spectra.

The rate of the NH_2_ flips was assessed at 30 °C from the exchange cross-peak intensities in 300 ms NOESY spectra, assuming that the equilibrium constant equals unity and equal on- and off-rate constants. Under these conditions, the ratio of cross and diagonal peak intensities would have the following form:(1)$$I_{\mathrm{cross}}/I_{\mathrm{diag}}=\frac{1-e^{-k_{\mathrm{ex}}t}}{1+e^{-k_{\mathrm{ex}}t}}$$
where k_ex_ = k_1_+k_−1_ = 2k, and t equals the mixing time in the NOESY experiment [44].

### 3.3. Synthesis

The initial amides of pyridine carboxylic acids **1**–**7** were used as commercial (Sigma-Aldrich, Saint Louis, MO, USA) reagents without additional purification.

To 8.5 mmol of 1,3-propanesultone, a solution of 8.5 mmol of amide **1**–**7** was added with stirring (Figure 1). The mixture was stirred at room temperature or was heated under reflux for 4 h. The next day, the solvent was evaporated, and the methanol or ethanol was added. The formed crystals were filtered off. Spin multiplicities were described as s (singlet), d (doublet), t (triplet), q (quartet), or m (multiplet).

*3-(2-(Carbamoylpyridinium-1-yl)propane-1-sulfonate* (**8**). The reaction was performed in 5 mL of methanol. Obtained 0.5 g (24% yield) sulfonate, m.p. 187-188 °C. IR (solid, ν/cm^−1^): 1706 (C=O), 1602 (C=C _pyridine_), 1167, 1146, 1171 (SO_3_). ^1^H-NMR (300.1 MHz, D_2_O, ppm, *J*/Hz): δ 2.41–2.58 (m, 2H, C-**CH_2_**-C), 2.92–3.06 (m, 2H, CH_2_SO_3_), 4.76–4.91 (m, 2H, N-CH_2_), 8.38 (d, 2H, ^3^*J* = 6.4, H4, H6), 9.07 (d, 2H, ^3^*J* = 6.4, H3, H6); ^13^C-NMR (75.5 MHz, D_2_O, ppm): δ 26.22, 47.10, 60.36, 126.63, 148.80, 166.81; Anal. calcd. for C_9_H_12_N_2_O_4_S: C, 44.26; H, 4.92; N, 11.48; S, 13.11; Found: C, 43.98; H, 5.01; N, 11.30; S, 12.89.*3-(2-(Methylcarbamoyl) pyridinium-1-yl)propane-1-sulfonate* (**9**). Reaction was performed in 5 mL of methanol. Recrystallized from ethanol-water, 60:1. Obtained 0.08 g (36% yield) sulfonate, m.p. 237–240 °C. IR (solid, ν/cm^−1^): 1677 (C=O), 1618 (C=C _pyridine_), 1210, 1150, 1036 (SO_3_). ^1^H-NMR (300.1 MHz, D_2_O, ppm, *J*/Hz): δ 2.35–2.49 (m, 2H, C-**CH_2_**-C), 2.97–3.03 (m, 2H, CH_2_SO_3_), 3.00 (s, 3H, CH_3_) 4.84 (t, 2H, ^3^*J* = 7.7, N-CH_2_), 8.66–8.70 (m, 2H, H4, H5), 8.65 (t, H, ^3^*J* = 7.5, H6), 9.01–9.04 (m, 1H, H3); ^13^C-NMR (75.5 MHz, D_2_O, ppm): δ 26.33, 26.62, 47.34, 58.11, 127.95, 129.53, 132.51, 146.78, 147.38, 162.07; Anal. calcd. for C_10_H_14_N_2_O_4_S: C, 44.93; H, 5.65; N, 10.48; S, 11.99; Found: C, 45.28; H, 5.70; N, 10.27; S, 12.38.*3-(3-Carbamoylpyridinium-1-yl)propane-1-sulfonate* (**10**). (a) Reaction was performed in 5 mL of methanol. Recrystallized from ethanol-water, 2:1. Obtained 1.86 g (89% yield) sulfonate, m.p. > 300 °C (276–280 °C, ethanol-water, 2:1 [27]). IR (solid, ν/cm^−1^): 1685 (C=O), 1654 (C=C _pyridine_), 1211, 1156, 1140 (SO_3_). ^1^H-NMR (300.1 MHz, D_2_O, ppm, *J*/Hz): δ 2.36–2.46 (m, 2H, C-**CH_2_**-C), 2.93 (t, 2H, ^3^*J* = 7.1 CH_2_SO_3_), 4.77 (t, 2H, ^3^*J* = 7.3, N-CH_2_), 8.12 (t, 1H, ^3^*J* = 7.0, H5), 8.82 (d, 1H, ^3^*J* = 8.1, H6), 8.98 (d, 1H, ^3^*J* = 6.1, H4), 9.28 (s, 1H, H2); ^13^C-NMR (75.5 MHz, D_2_O, ppm): δ 26.08, 47.00, 60.48, 128.55, 134.11, 144.30, 144.49, 146.74, 165.79; Anal. calcd. for C_9_H_12_N_2_O_4_S: C, 44.26; H, 4.92; N, 11.48; S, 13.11; Found: C, 44.18; H, 4.96; N, 11.49; S, 13.06. (b) Reaction was performed in 5 mL of methanol. The mixture was heated under reflux for 4 h. Obtained 1.67 g (80% yield) sulfonate. The parameters of the ^1^H, ^13^C NMR, and FT-IR spectra of compounds obtained at different temperatures (a and b) have the same values.*3-(3-(Methylcarbamoyl)pyridinium-1-yl)propane-1-sulfonate* (**11**). Reaction was performed in 5 mL of methanol. Recrystallized from ethanol–water, 5:1. Obtained 1.84 g (97% yield) sulfonate, m.p. 280–282 °C. IR (solid, ν/cm^1^): 1661, 1631 (C=O), 1593 (C=C _pyridine_), 1202, 1174, 1150 (SO_3_). ^1^H-NMR (300.1 MHz, D_2_O, ppm, *J*/Hz): δ 2.32–2.48 (m, 2H, C-**CH_2_**-C), 2.84–2.96 (m, 2H, CH_3_), 2.96 (s, 3H, CH_2_SO_3_), 4.76 (t, 2H, ^3^*J* = 7.8, N-CH_2_), 8.11 (t, 1H, ^3^*J* = 6.8, H5), 8.76 (d, 1H, ^3^*J* = 7.8, H6), 8.96 (d, 1H, ^3^*J* = 6.6, H4); 9.22 (s, 1H, H2); ^13^C-NMR (75.5 MHz, D_2_O, ppm): δ 26.03, 26.61, 46.97, 60.46, 128.54, 134.69, 143.87, 144.07, 146.45, 164.16; Anal. calcd. for C_10_H_14_N_2_O_4_S: C, 46.50; H, 5.46; N, 10.84; S, 12.41; Found: C, 46.09; H, 5.45; N, 10.66; S, 12.43.*3-(3-(Diethylcarbomoyl)pyridinium-1-yl)propane-1-sulfonate* (**12**). The reaction was performed in 5 mL of methanol. Recrystallized from acetonitrile–ethanol, 10: 1. Obtained 1.52 g (59% yield) sulfonate, m.p. 197–199 °C. IR (solid, ν/cm^−1^): 1620 (C=O), 1201, 1182, 1037 (SO_3_). ^1^H-NMR (300.1 MHz, D_2_O, ppm, *J*/Hz):δ 1.07 (t, 3H, ^3^*J* = 7.2, CH_2_-**CH_3_**), 1.17 (t, 3H, ^3^*J* = 7.2, CH_2_-**CH_3_**), 2.36–2.46 (m, 2H, C-**CH_2_**-C), 2.86–2.96 (m, 2H, CH_2_SO_3_), 3.23 (q, 2H, ^3^*J* = 7.4, **CH_2_**-CH_3_), 3.50 (q, 2H, ^3^*J* = 7.4, **CH_2_**-CH_3_), 4.76 (t, 2H, ^3^*J* = 7.5, N-CH_2_), 8.12 (t, 1H, ^3^*J* = 7.1, H5), 8.09–8.15 (m, 1H, H6), 8.53–8.58 (m, 1H, H4), 9.04 (s, 1H, H2); ^13^C-NMR (75.5 MHz, D_2_O, ppm): δ 11.76, 13.02, 26.18, 40.72, 44.42, 47.00, 60.49, 129.06, 136.49, 142.31, 143.55, 145.61, 165.37. Anal. calcd. for C_13_H_20_N_2_O_4_S∙H_2_O: C, 49.04; H, 6.96; N, 8.80; S, 10.07; Found: C, 48.54; H, 6.71; N, 8.34.*3-(4-Carbamoylpyridinium-1-yl)propane-1-sulfonate* (**13**). (a) Reaction was performed in 5 mL of methanol. Recrystallized from ethanol–water, 5:2. Obtained 1.20 g (58% yield) sulfonate, m.p. 283–284 °C. IR (solid, ν/cm^−1^): 1692 (C=O), 1626 (C=C _pyridine_), 1199, 1164, 1130 (SO_3_). ^1^H-NMR (300.1 MHz, D_2_O, ppm, *J*/Hz): δ 2.49–2.51 (m, 2H, C-CH_2_-C), 3.01 (t, 2H, ^3^*J* = 7.3, CH_2_SO_3_), 4.85 (t, 2H, ^3^*J* = 7.5, N-CH_2_), 8.38 (d, 2H, ^3^*J* = 6.3, H3, H5), 9.06 (d, 2H, ^3^*J* = 6.3, H2, H6); ^13^C-NMR (75.5 MHz, D_2_O, ppm): δ 26.22, 47.12, 60.34, 126, 145, 166.82, 34.99, 92.23, 111.47, 113.98, 119.10, 124.92, 139.13, 151.75, 159.51. Anal. calcd. for C_9_H_12_N_2_O_4_S: C, 44.25; H, 4.95; N, 11.47; S, 13.13; Found (%): C, 44.01; H, 4.90; N, 11.25; S, 13.04. (b) Reaction was performed in 5 mL of methanol. The mixture was heated under reflux for 4 h. Obtained 1.30 g (67% yield) sulfonate. The parameters of the ^1^H, ^13^C NMR, and FT-IR spectra of compounds obtained at different temperatures have the same values.*3-(4-(Methylcarbamoyl)pyridinium-1-yl)propane-1-sulfonate* (**14**). The reaction was performed in 5 mL of methanol. Obtained 1.66 g (76% yield) sulfonate, m.p. > 300 °C. IR (solid, ν/cm^−1^): 1665 (C=O), 1645 (C=C _pyridine_), 1178, 1143, 1036 (SO_3_). ^1^H-NMR (300.1 MHz, D_2_O, ppm, *J*/Hz): δ 2.33–2.44 (m, 2H, C-CH_2_-C), 2.83–2.95 (m, 2H, CH_2_SO_3_), 2.88 (s, 3H, CH_3_), 4.75 (t, 2H, ^3^*J* = 7.6, N-CH_2_), 8.24 (d, 2H, ^3^*J* = 6.1, H3, H5), 8.96 (d, 2H, ^3^*J* = 6.1, H2, H6); ^13^C-NMR (75.5 MHz, D_2_O, ppm): δ 26.10, 26.64, 46.96, 60.13, 126.22, 145.64, 149.09, 165.04. Anal. calcd. for C_10_H_14_N_2_O_4_S: C, 46.50; H, 5.46; N, 10.85; S, 12.41; Found: C, 46.15; H, 5.31; N, 10.53.*2-Carbamoylpyridin-1-ium 3-methoxypropane-1-sulfonate* (**15**).(a) Reaction was performed in 5 mL of methanol. Mixture was heated under reflux for 4 h. Obtained 1.52 g (65% yield) complex (**15**), m.p. 153–155 °C (benzene-acetonitrile, 1:2). IR (solid, ν, cm^−1^): 1707 (C=O), 1602 (C=C _pyridine_), 1179, 1124, 1035, (SO_3_). ^1^H-NMR (300.1 MHz, D_2_O, ppm, *J*/Hz): δ 1.88 (m, 2H, C-**CH_2_**-C), 2.81 (t, 2H, ^3^*J* = 7.1 CH_2_SO_3_), 3.46 (t, 2H, ^3^*J* = 7.3, CH_3_O-**CH_2_**), 3.23 (s, 3H, **CH_3_O**-CH_2_), 8.08 (t, 1H, ^3^*J* = 7.0, H5), 8.77 (d, 1H, ^3^*J* = 8.1, H6), 8.35 (d, 1H, ^3^*J* = 6.1, H4), 8.58 (t, 1H, ^3^*J* = 7.0, H5); ^13^C-NMR (75.5 MHz, D_2_O, ppm): δ 25.55, 26.56, 58.18, 60.36, 71.20 122.79, 127.52, 140.35, 147.68, 148.77, 163.63; Anal. calcd. for C_10_H_16_N_2_O_5_S: C, 43.47; H, 5.84; N, 10.14; S, 11.60; O, 28.95; Found: C, 42.85; H, 6.54; N, 9.09; S, 10.40; O, 31.13.(b) Reaction was performed in 5 mL of methanol-d_4_. Mixture was heated under reflux for 4 h. Obtained 2.21 g (93% yield) complex (**15-d**), m.p. 149-153 °C (benzene-acetonitrile, 1: 2). IR (solid, ν, sm^−1^): 1687 s (C=O), 1609 (C=C _pyridine_), 1184, 1128, 1035, (SO_3_). ^1^H-NMR (300.1 MHz, DMSO-*d*_6_, ppm, *J*/Hz): δ 1.75–1.87 (m, 2H, C-**CH_2_**-C), 2.56-2.66 (m, 2H, CH_2_SO_3_), 3.33 (t, 2H, ^3^*J* = 7.3, CD_3_O-**CH_2_**), 7.82 (m, 1H, H5), 8.71 (m, 1H, H6), 8.03 (m, 1H, H4), 8.29 (m, 1H, H5); ^13^C-NMR (75.5 MHz, DMSO-*d*_6_, ppm): δ 25.58, 48.79, 71.11, 123.43, 127.99, 141.45, 147.12, 148.02, 164.42.*2-(Methylcarbamoyl)pyridin-1-ium 3-methoxypropane-1-sulfonate* (**16**). Reaction was performed in 5 mL of methanol. Mixture was heated under reflux for 4 h. Obtained 2.00 g (81% yield) oily complex, m.p. 60–64 °C. IR (solid, ν/cm^−1^): 1679 s (C=O), 1604 (C=C _pyridine_), 1212, 1108, 1032, (SO_3_). ^1^H-NMR (300.1 MHz, D_2_O, ppm, *J*/Hz): δ 1.92 (m, 2H, C-**CH_2_**-C), 2.85 (t, 2H, ^3^*J* = 7.1 CH_2_SO_3_), 3.50 (t, 2H, ^3^*J* = 7.3, CH_3_O-**CH_2_**), 3.26 (s, 3H, **CH_3_O**-CH_2_), 2.95 (s, 3H, HN**CH_3_**), 8.12 (t, 1H, ^3^*J* = 7.0, H5), 8.81 (d, 1H, ^3^*J* = 8.1, H6), 8.34 (d, 1H, ^3^*J* = 6.1, H4), 8.63 (t, 1H, ^3^*J* = 7.0, H5); ^13^C-NMR (75.5 MHz, D_2_O, ppm): δ 24.03, 26.90, 47.78, 57.61, 70.47, 125.06, 129.57, 143.21, 146.96, 162.39. C_11_H_18_N_2_O_5_S HR-MS (ESI) calcd. for C_6_H_6_N_2_OCH_3_^+^: 137.16 [M + H^+^]^+^, found 137.07; calculated for C_4_H_9_O_4_S^+^: 153.17, found 153.02 [M + H^+^]^+^.

### 3.4. Calculation Details

All quantum-chemical calculations were carried out using GAUSSIAN 09 RevC software [45]. The popular hybrid functional (M052X [46]) with empirical dispersion [47] combined with the basis set TZVP [48] was used throughout the work for geometry optimization and frequency calculations. This calculation method was chosen by the results of comparable experimental and computational chemical shifts in the NMR spectra. In this case, the correlation index is more than 99% (see details in Appendix A, ^1^H and ^13^C NMR chemical shifts).

Geometrical parameters of possible reagents and products were carried out. Estimation of the value of the activation barrier was carried out by searching for transition states. Hessian matrix was positively denoted as the total energy, then the stationary point attained a local minimum of the PES. If the Hessian had the only negative eigenvalue (imaginary frequency), then the stationary point corresponded to a saddle point (transition state, TS). The search for transient states was carried out in two stages: a relaxed scan by the reaction coordinate to find the maximum point (transition state) on the PES, followed by the standard procedure for localizing the stationary maximum point. The reaction products and the pre-reaction complex are predicted by the IRC procedure, for which 30 points on the PES in both directions were selected. Thermodynamic parameters were calculated in the gas-phase approximation at 298 K and also by simulating the experimental conditions: methanol as the solvent (polarized continuum model [49]) at 298 K and a pressure of 1 atm. Pre-reaction complexes of disclosure reactions sulton cycle were created based on analysis of ESP electrostatic potential maps that were calculated in the MultiWFN program version 3.8 [50]. ESP maps were rendered using VMD [51]. Chemical shifts were calculated using the Continuous Set of Gauge Transformations (CSGT) method [52,53,54].

### 3.5. X-ray Diffraction Studies

Single crystal X-ray studies of products **9**, **12**, and **14** were carried out in the Center for Molecule Composition Studies of A.N. Nesmyanov Institute of Elementoorganic Chemistry, Russian Academy of Sciences (Moscow; Russia) with Bruker APEX II diffractometer. X-ray datasets for products **10**, **11**, and **13** were collected in the Kurchatov Centre for Synchrotron Radiation and Nanotechnology using the ‘Belok’ beamline. All structures were solved with the ShelXT version 2014/4 [55] program and refined with the ShelXL [56] program. Molecular graphics were drawn using the OLEX2 version 1.3 [57] program.

CCDC 2125076–2125081 contains the supplementary crystallographic data for **9**–**14** (see Appendix A). These data can be obtained free of charge from The Cambridge Crystallographic Data Centre via https://www.ccdc.cam.ac.uk/structures, accessed on 25 September 2022.

### 3.6. Cell Culture

RAW 264.7, a murine macrophage cell line, was kindly provided by laboratory of the mechanism of chemical carcinogenesis, N.N. Blokhina Russian Cancer Research Center, was grown in high-glucose Dulbecco’s modified Eagle’s medium (DMEM), supplemented with 10% fetal bovine serum (FBS) and 1% antibiotics (penicillin and streptomycin) in a humidified incubator flushed continuously at 37 °C with 5% CO_2_. The culture medium was refreshed every 3 days.

#### 3.6.1. Cell Viability Assay

The cell viability was measured according to the cell ability of tetrazolium dye reduction. The Thiazolyl Blue Tetrazolium Bromide solution was added to the culture medium at a final concentration of 0.5 mg/mL, and the cells were left in darkness for 4 h at 37 °C. After that, the solution was removed, and formazan crystals were dissolved in dimethyl sulfoxide. The formazan absorption at 492 nm was measured using a Zenyth 1100 (Anthos Labtec Instruments, GmbH, Salzburg, Austria) microplate reader. The experiments were carried out in triplicate and repeated three times.

#### 3.6.2. Quantitative Real-Time PCR Analysis

After overnight culture in a 6-well plate (25 × 104 cells/well, 2000 μL medium/well), the cells were treated with LPS (*Escherichia coli* L4130 0111:B4, Sigma-Aldrich, Saint Louis, MO, USA) at a concentration of 10 μg/mL and were incubated with testing compounds (0.001, 0.01, 0.1 and 0.5 mg/mL) for an additional 6 h. The anti-inflammatory effect of compounds was compared with the effect caused by LPS within 24 h. After 24 h of total incubation of cells with LPS and 6 h of incubation with compounds, the cells were scraped to analyze the expression of target genes.

Analysis of the level of gene expression was carried out using real-time reverse transcription-polymerase chain reaction (PCR).

Total RNA was isolated from cells using the RNA-Extran kit (Synthol, Moscow, Russia, Ex-515-50) following the manufacturer’s protocol. The corresponding cDNA (cDNA) was obtained during a reverse transcription reaction using the OT-1 reagent kit (Synthol, Moscow, Russia) and a PTC (100) programmable device (MJ Research Inc., Deltona, Florida, USA). The reaction program included incubation of the mixture for 10 min at 25 °C, 30 min at 39 °C, and 5 min at 92 °C, followed by cooling to 4 °C.

Real-time PCR was performed on a CFX96 Touch™ Real-Time PCR Detection System (Bio-Rad Laboratories, Hercules, CA, USA) in the presence of Eva Green fluorescent dye (Synthol, Moscow, Russia). The amplification program included 3 min at 95 °C and 40 cycles (10 s 95 °C, 10 s 57 °C, 30 s 72 °C). The composition of the reaction mixture (25 μL): 1 × PCR buffer; 8 mM MgCl_2_; 0.25 mM mixture of deoxynucleotide triphosphates (dNTPs); 0.2 units Taq polymerases; 200 nM forward and reverse primers (Synthol, Moscow, Russia). The primers were selected using the Primer-BLAST (NCBI) datebase and are presented in Table 6. To determine the change in the expression level, the threshold cycle calculation method was used. Samples were normalized to the amount of mRNA of the Rpl27 gene.

## 4. Conclusions

The position of the substituent in the aromatic ring of pyridine carboxamides based on sulfobetaines affects not only the yield of the target product, but also the reaction mechanism. Following the results of quantum-chemical calculations, the interaction of the studied substrates with 1,3-propanesultone is subject to charge control. According to IR and NMR spectroscopy data, in substrates **1** and **2** an intramolecular hydrogen bond is formed between the hydrogen atom of the amide group in the *ortho*-position and the endocyclic nitrogen atom of the heteroaromatic fragment. As a result, the yield of the reaction product in the pre-reaction complex decreases to 20–40% as compared to the *meta*- and *para*-substituted analogs (62–97%).

In addition to the charge factor, the high yield (89–97%) of *meta*-substituted derivatives **10** and **11** can be influenced by additional activation of the “substrate–1,3-propanesultone” transition state due to the formation of a hydrogen bond with the participation of the endocyclic oxygen atom of 1,3-propanesultone and a hydrogen atom of the amide group.

Steric factors or lack of internal hydrogen bond formation also influence the yield of the targeted sulfobetaines because, with the tertiary carboxamide analog of nicotinamide (substrate **5**), global yield decreases from 10 to 24% compared to nicotinamide **3** or *N*-methyl analog **4**, respectively.

The combination of experimental data, the thermodynamic parameters of the cations, and charge control explain the difference in the reactivity of *ortho*-substituted **1** and **2** at high temperature compared to compounds **3** and **6**.

The results of in silico analysis and preliminary in vitro analysis showed low toxicity of these compounds and possible anti-inflammatory activity. Therefore, we may consider these compounds as potential therapeutic candidates for further research.

## Data Availability

All spectra and XRD data are available from the authors.

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
