# Peer review of "Pyridine Carboxamides Based on Sulfobetaines: Design, Reactivity, and Biological Activity"

_molecules, 2022, doi:10.3390/molecules27217542_

Round 1
Reviewer 1 Report
The manuscript proposed by D. Kramarova and collaborators untitled « Designing Sulfobetaines on the Base of Pyridine Amides : Reactivity, Structure and Biological Activity» and submitted to Molecules (molecules-1964220-peer-review-v1), described the synthesis, chemical and physicochemical analyses (XRay and IR) of a series of seven sulfobetaines obtained by the reaction of the corresponding pyridine carboxamides (eg nicotinamide and analogs) with 1,3-propane sultone. With these collected experimental data, completed with a theoritical study on the electronic properties of the pyridine carboxamides, the authors implied a mechanism for the reaction. They finally proposed to predict with in silico approach, some potential biological activities and the compounds were also tested in vitro to evaluate their potential toxicity on four cell lines, and their anti-inflammatory response on the RAW 264.7 macrophage cell line.
If the synthetic pathway to prepare the seven compounds is a classical one, some of the targeted sulfobetaines are new such as the sylfobetaines obtained from ortho or para analogs of nicotinamide (compounds 1-2 and 6-7). The authors have collected serious experimental physicochemical data of starting materials and products (NMR, XR and IR) and driven quantum-chemical calculations for a better comprehension of the reactivity of these pyridine carboxamides towards sultone to conclude on a plausible mechanism for this alkylation reaction. Some interesting anti-inflammatory property has been observed with the sulfobetaine derived from N,N-diethyl nicotinamide, which could open the way to the developement of novel sulfobetaines as therapeutic agents (Ben-Sasson et al ; WO2009136396 A2 2009-11-12).
Taking into account that some of the sulfobetaines described in the paper are novel compounds and that the experimental or in silico studies remain of interest, the reviewer accept this work for publication in Molecules, with some corrections and complementary informations. The authors also need to check their writing - in several places the writing expression is awkward or confusing.
General comments :
11. In the title and along the manuscript the term pyridine carboxamide (derivatives of pyridine carboxylic acid) could be prefered to pyridine amide (derivatives of amino pyridine !) .
22. Abstract could be simplified, for example : delete the sentence (line 24-25 : The products…). Fom line 25 to 32, simplify : « On the basis of spectroscopic, X-Ray and quantum-chemical calculation data, a model of the transition-state, as well as a mechanism for this alkylation reaction of pyridine carboxamides with sultone were proposed, to explain the higher yields obtained with the nicotinamide and its N-methyl analog compared to ortho or meta parents. »
33. Along the text change « pyrimidine » with « pyridine » (lines 101, 140, et ….).
44. The conclusion must be revised, the first § is confusing.
Specific comments in the text:
11. Line 51 : In homotaurin the terminal nitrogen of the sulfobetaine is not involved in a pyridine ring, so homotaurin fragment is not right !
22. In scheme 1, add the reaction time, and delete CH3CN, because there is no discussion about the reaction with acetonitrile and in experimental section all the syntheses (8 to 14) are described in refluxing methanol ? In this title « …compounds 8-14 »
43. In table 1 : what does « id » mean, is it the number ?
44. Lines 66-67 : The authors could precise if the low yields observed with compounds 8 an 9 are due to degradation or to low consumption of the starting materials ?
5. Lines 74-75 : Give a reference for this affirmation.
6.5. Line 76 : The carbamoyl pyridinyl fragment of compound 13 also seems to be coplanar and is not discussed ?
7.6. Lines 100-106 : The authors must explain with more precisions the ESP value around the pyridine nitrogen atoms of compounds 1 and 2 (associated with high ESP values around the CO group ) ? The expression « reduced in potential dropping » is not clear ?
8.7. Line 110 : ….in sultone AND pyridine substrates 1-7 ?
88.. In table 2, what is the symbolassociated with H and G, D ?
19.. Line 158, rewrite : « These results are consistent with the lowest yields obtained with compounds 8 and 9 ».
110. Lines 160-166 : This paragraph is not clear : thermodynamic and activation parameters of all the compounds are compared with their yields, and only compound 12 doesn’t match, so it is concluded that steric factors are suggested for this compound 12?
111. In Figure 4, it is suggested to delete compound 12 for more clarity and to identify the carbamoyl NH2 with a bullet and the carbamoyl NHMe with a square (for exemple).
112. Line 220, rewrite : ….steric hindrance of the pyridine nitrogen atom with amide hydrogen atom of the carbamoyl group….
113. § 2.3.2 : para-substituted compounds 13 and 14 are not discussed in this paragraphe
114. Lines 249-254 : IR spectra are done in water ? Water must probably interact with the amide bond (solvatation) as well as the SO3-, and block potential intramolecular bond formation ? So the IR experiments could be preferentially driven in aprotic solvents to give the expected information?
115. § 2.3.3 Please argument your unexpected results with some references describing alcoholysis of sultone generally done in basic conditions (using NaH, etc…) ; The same experience could be done in refluxing acetonitrile?.
116. In scheme 3, detailed reaction conditions (temperature, time of reaction….)
117. Figure 9 must be moved in SI.
118. For in vitro studies, toxicity on cancer cell lines (HaCat ???) have been evaluated, why not on RAW264.7 cells line, which was later used to evaluate anti-inflammatory potential of compound 12 ?
219. Lines 609-612 :rewrite for example; « Steric factors or lack of internal hydrogen bond formation, also influence the yield of the targeted sulfobetaines because with the tertiary carboxamide analog of nicotinamide (substrate 5) , global yield decrease from 10 to 24 % compared to nicotinamide 3 or N-methyl analog 4, respectively…. »
Specific comments on the experimental section:
1.1. Lines 416-420 : Duration of the reaction at room temperature is 4 hours or several days ?
2.2. Compound 8 : 1H NMR description doesn’t match with an ortho substituted pyridine as well as with 1H NMR of compound 8 in SI ?
33. Compounds 14 an 15 1H NMR description : H3 is missing ?
44. Lines 487, 495 and 502 : traduction in english language
Author Response
The authors thank the reviewers for the careful reading of the manuscript, gratefully accept all comments and suggestions for improving the manuscript. The authors also apologize for technical typing mistakes and resulting misunderstandings.
Response to Reviewer 1 Comments
Responses to Points 6, 7, 13, 15,16 and 21 - Please see attachment
Point 1: In the title and along the manuscript the term pyridine carboxamide (derivatives of pyridine carboxylic acid) could be prefered to pyridine amide (derivatives of amino pyridine
Response 1: The necessary changes have been made throughout the text of the article. Additionally, the title of the article was corrected: «Pyridine Carboxamides Вased on Sulfobetaines: Design, Reactivity, and Biological Activity».
Point 2: Abstract could be simplified, for example : delete the sentence (line 24-25 : The products…). Fom line 25 to 32, simplify : « On the basis of spectroscopic, X-Ray and quantum-chemical calculation data, a model of the transition-state, as well as a mechanism for this alkylation reaction of pyridine carboxamides with sultone were proposed, to explain the higher yields obtained with the nicotinamide and its N-methyl analog compared to ortho or meta parents.
Response 2: The proposed changes have been taken into account and included in the relevant sections of the manuscript.
Point 3: Along the text change « pyrimidine » with « pyridine » (lines 101, 140, et ….).
Response 3: The proposed changes have been taken into account and included in the relevant sections of the manuscript
Point 4: The conclusion must be revised, the first § is confusing
Response 4: The proposed changes have been taken into account and included in the relevant sections of the manuscript.
The position of the substituent in the aromatic ring of pyridine carboxamides based on sulfobetaines affects not only the yield of the target product, but also the reaction mechanism.
Point 5: Line 51 : In homotaurin the terminal nitrogen of the sulfobetaine is not involved in a pyridine ring, so homotaurin fragment is not right !
Response 5: The following correction has been made: For the synthesis of pyridinecarboxylic acids, studied in this work, containing a homotaurin «tail», we used the previously proposed method [23]
Point 8: Lines 66-67 : The authors could precise if the low yields observed with compounds 8 an 9 are due to degradation or to low consumption of the starting materials ?
Response 8: Thank you so much for your question. We are not inclined to associate the low yield of compounds 8 and 9 with either degradation or low availability of starting substances. The most probable cause is the shielding of the reaction center by a substituent in the ortho position. The corresponding conclusion based on experimental data is made at the end of section 2.1.1:«We conclude that steric hindrance of the pyridine nitrogen atom with amide hydrogen atom of the carbamoyl group is the second point (along with low ESP) which decreases the yield of the reaction».
Point 9: Lines 74-75 : Give a reference for this affirmation.
Response 9: Thanks for the comment. The reference was placed into corresponding paragraph “The values of bond length and angles of the abovementioned compounds are very close to organic compounds with similar fragments fragments [Allen, F.H.; Kennard, O.; Watson, D.G.; Brammer, L.; Orpen, A.G.; Taylor, R. Tables of Bond Lengths Determined by X-Ray and Neutron Diffraction. Part 1. Bond Lengths in Organic Compounds. J. Chem. Soc., Perkin Trans. 2 1987, S1–S19.]”
Point 10: Line 76 : The carbamoyl pyridinyl fragment of compound 13 also seems to be coplanar and is not discussed ?
Response 10: Yes, in 13 these fragments are also coplanar. The text was changed accordingly. The substituted pyridine ring and amide CONR1R2 moieties in molecules 10, 11 and 13 are coplanar
Point 11: Lines 100-106 : The authors must explain with more precisions the ESP value around the pyridine nitrogen atoms of compounds 1 and 2 (associated with high ESP values around the CO group ) ? The expression « reduced in potential dropping » is not clear ?
Response 11: We are fixed on “Changing the position of the ring substituent from the para- to the meta- and ortho- positions leads to potential dropping from –60.1 kJmol–1 (compound 1, Figure 2) to –138.7 kJmol–1 for the meta- position (compound 3) and –147.0 kJmol–1 for para-substitution (compound 6)”.
Point 12: Line 110 : ….in sultone AND pyridine substrates 1-7 ?
Response 12: Necessary corrections have been made: ESP extrema corresponding to regions exhibiting pronounced electron-withdrawing properties are marked with red spheres in sultone and pyridine substrates 1–7.
Point 14: Line 158, rewrite : « These results are consistent with the lowest yields obtained with compounds 8 and 9 ».
Response 14: The proposed changes have been taken into account and included in the relevant sections of the manuscript.
Point 17: Line 220, rewrite : ….steric hindrance of the pyridine nitrogen atom with amide hydrogen atom of the carbamoyl group
Response 17: The proposed changes have been taken into account and included in the relevant sections of the manuscript.
Point 18: § 2.3.2 : para-substituted compounds 13 and 14 are not discussed in this paragraphe
Response 18: Thank you for your question. The yields of compounds 13 and 14 are discussed in this section through comparison with the initial compounds 6 and 7. Clarifications are made in the corresponding section: "All para-derivatives (compounds 6, 7) and meta-diethylamide pyridine (compound 5) show moderate yields in sultone alkylation (products 13, 14), while meta-amide (compound 3) and meta-methylamide (compound 4) demonstrate sufficiently higher yields (products 10, 11) (Table 1)».
Point 19: Lines 249-254 : IR spectra are done in water ? Water must probably interact with the amide bond (solvatation) as well as the SO3-, and block potential intramolecular bond formation ? So the IR experiments could be preferentially driven in aprotic solvents to give the expected information?
Response 19: Thanks for the question. Indeed, the spectra of the studied compounds were obtained in water. We are well aware that the solvent used, due to the reasons listed by the reviewer, is not ideal for studying the formation of both intra- and intermolecular bonds. Nevertheless, the reaction we investigated was carried out in an aqueous-alcohol medium due to the low solubility of both the compounds studied and the reaction products in aprotic solvents. In this regard, we were also interested in the effect of solvation effects on the mechanism of the reaction, as well as on the yield of products.
Point 20: § 2.3.3 Please argument your unexpected results with some references describing alcoholysis of sultone generally done in basic conditions (using NaH, etc…) ; The same experience could be done in refluxing acetonitrile?
Response 20: А) The following has been added to the specified section: "The disclosure of the propanesultone cycle under similar conditions is described in a few reports deal with the alcoholysis of sultones [Helberger, J. H.; Heyden, J. R.; Winter, H. Liebigs Ann. Chem. 1954, 586, 147; D. Enders, W. Harnying / Asymmetric synthesis of α,γ-substituted γ-alkoxy methyl sulfonatesvia diastereospecific ring-opening of sultones // ARKIVOC. 2004, (ii) , p. 181-188. https://doi.org/10.3998/ark.5550190.0005.212 and the works cited in it]
В) An experiment similar to the one described in section 2.3.3 was also performed in the refluxing acetonitrile. In this case, we were unable to obtain compounds of a similar structure to compounds 15, 16.
Point 22: Figure 9 must be moved in SI.
Response 22: The proposed changes have been taken into account and included in the relevant sections of the manuscript.
Point 23: For in vitro studies, toxicity on cancer cell lines (HaCat ???) have been evaluated, why not on RAW264.7 cells line, which was later used to evaluate anti-inflammatory potential of compound 12 ?
Response 23:
Cell line HaCaT - immortalized human keratinocyte line Cellosaurus ID CVCL_0038
The first step in the study of biological activity was to determine the possible toxicity of new compounds. We did a toxicity study on a panel of different cells. To study the anti-inflammatory activity, was been taken a classical cell line. Of course, we tested the cytotoxicity of the new compounds on Raw 264.7 cells. But since the maximum concentration for the study of anti-inflammatory activity was 4.8 times lower than the minimum toxicity, we made the decision not to include these data in the results. If necessary, we will include them.
Point 24: Lines 609-612 :rewrite for example; « Steric factors or lack of internal hydrogen bond formation, also influence the yield of the targeted sulfobetaines because with the tertiary carboxamide analog of nicotinamide (substrate 5) , global yield decrease from 10 to 24 % compared to nicotinamide 3 or N-methyl analog 4, respectively…. »
Response 24: The proposed changes have been taken into account and included in the relevant sections of the manuscript.
Point 25: Lines 416-420 : Duration of the reaction at room temperature is 4 hours or several days
Response 25: - Technical error. Fixed:
The next day, the solvent was evaporated, and the methanol or ethanol was added.
Point 26: Compound 8 : 1H NMR description doesn’t match with an ortho substituted pyridine as well as with 1H NMR of compound 8 in SI ?
Response 26: 1Н NMR (D2O, δ, ppm, J/Hz): 2.41-2.58 (m, 2H, C-CH2-C), 2.92-3.06 (m, 2H, CH2SO3), 4.76–4.91 (m, 2H, N-CH2), 8.01 (d, 1H, 3J 6.4, H3), 8.15 (d, 1H, 3J 6.4, H6), 8.39 (t, 1H, 3J 6.4, H4), 8.73 (d, 2H, 3J 6.4, H3, H6).
13С NMR (D2O, δ, ppm): 24.01, 47.77, 57.61, 70.48, 124.89, 129.37, 143.68, 146.20, 163.14.
Point 27: Compounds 14 an 15 1H NMR description : H3 is missing ?
Response 27: Thanks a lot for the comment. It allowed us to find some technical typos. Numbering begins with the nitrogen atom (heteroatom). Signal attribution and chemical shifts are verified. Fixed technical errors. Adjustments have been made.
Comp 14. 1Н NMR (D2O, δ, ppm, J/Hz): 2.33–2.44 (m, 2H, C-СН2-С), 2.83–2.95 (m, 2H, CH2SO3), 2.88 (s, 3Н, NCH3), 4.75 (t, 2H, 3J 7.6, NCH2), 8.24 (d, 2Н, 3J 6.1, H3, H5), 8.96 (d, 2H, 3J 6.1, H2, H6).
13С NMR (D2O, δ, ppm): 26.10, 26.64, 46.96, 60.12, 126.21, 145.64, 149.07, 165.02.
Comp. 15. 1H NMR (D2O, δ, ppm, J/Hz): 1.88 (m, 2H, C-СН2-С), 2.81 (t, 2Н, 3J 7.1 CH2SO3), 3.46 (t, 2Н, 3J 7.3, CH3O-СН2), 3.23 (s, 3Н, CH3O-СН2), 8.08 (t, 1Н, 3J 7.0, H5), 8.77 (d, 1H, 3J 8.1, H6), 8.35 (d, 1Н, 3J 6.1, H4), 8.58 (t, 1Н, 3J 7.0, H5).
13C NMR (D2O, δ, ppm): 24.03, 47.78, 57.61, 70.47, 125.06, 128.37, 129.57, 143.21, 146.88, 162.39
Comp. 16. 1H NMR (D2O, δ, ppm, J/Hz): 1.88 (m, 2H, C-СН2-С), 2.81 (t, 2Н, 3J 7.1 CH2SO3), 3.46 (t, 2Н, 3J 7.3, CH3O-СН2), 3.23 (s, 3Н, CH3O-СН2), 7.71 (t, 1Н, 3J 7.0, H4), 8.12 (m, 2H, H2, H6), 8.65 (t, 1Н, 3J 7.0, H5).
13C NMR (D2O, δ, ppm): 25.55, 26.56, 48.74, 58.18, 71.20, 122.79, 127.52, 140.35, 147.68, 148.77, 163.63
Point 28: Lines 487, 495 and 502 : traduction in english language
Response 28: - Technical error. Fixed

Reviewer 2 Report
The research topic is interesting and a wide range of studies was conducted. Unfortunately, I cannot recommend this article for publishing in this form. I would like to ask the authors to improve the text, especially the introduction and results and discussion sections.
Title
I would recommend you change the title for more accuracy. For example Pyridinecarboxamide based sulfobetaines: design, reactivity, and biological activity.
Keywords
The keywords are very general and do not represent all the research and analysis which were done. It will be hard for future readers to find this paper.
Introduction
The introduction gives very limited information about the topic. Do not explain why the following compounds were chosen and the used technics. More information is required to give sufficient background. The aim of the study is unacceptable.
Result and discussion
I propose changing the enumeration.
2.1. Synthesis
2.2. X-Ray investigation
2.3. Quantum-chemistry calculations
2.4. Spectroscopic analysis
Ect.
In the synthesis description, the reference to Scheme 1 is needed as basic information about the reaction condition.
A better formulation of the synthesis result analysis is needed.
The sentence “The values of bond length and angles of the abovementioned compounds are very 74 close to organic compounds with similar fragments.” should be confirmed by examples
For IR spectra the deconvolution could give more information about how many conformers are in the mixture.
Notes on tables, diagrams, phrases and spelling
Line 27: “subrate”
Scheme 1. There is a lack of description of R1 and R2 groups.
Table 1. It would be useful to add the synthesis details. The table description is too general.
Figure 2. Sentences exclude each other:
“Surface local minima and maxima of ESP are represented as blue and red spheres, respectively. The local surface maxima with negative ESP values are meaningless and thus not shown as”
Table 2. Which solvent was used?
Figure 2. The ESP charge is measured in kJ/mol?
Author Response
The authors thank the reviewers for the careful reading of the manuscript, gratefully accept all comments and suggestions for improving the manuscript. The authors also apologize for technical typing mistakes and resulting misunderstandings.
Response to Reviewer 2 Comments
Point 1: I would recommend you change the title for more accuracy. For example Pyridinecarboxamide based sulfobetaines: design, reactivity, and biological activity
Response 1: Thank you for your valuable suggestion. The title of the article has been corrected in accordance with the proposal.
Point 2: The keywords are very general and do not represent all the research and analysis which were done. It will be hard for future readers to find this paper.
Response 2: Thank you for your valuable suggestion. Corrected: derivatives of pyridine carboxylic acids; sulfobetaines; mechanism of reaction, NMR and FT-IR spectroscopy; X-Ray, quantum-chemical calculations
Point 3: and Response 3 Please see attachement
Point 4: Result and discussion I propose changing the enumeration.
2.1. Synthesis
2.2. X-Ray investigation
2.3. Quantum-chemistry calculations
2.4. Spectroscopic analysis
Ect.
Response 4: Thank you for your valuable suggestion. The necessary changes have been made to the manuscript.
Point 5: In the synthesis description, the reference to Scheme 1 is needed as basic information about the reaction condition.
Response 5: 3.3. Synthesis
The initial amides of pyridine carboxylic acids 1–7 were used as commercial (Acros and Sigma-Aldrich) reagents without additional purification. To 8.5 mmol of 1,3-propanesultone, a solution of 8.5 mmol of amide 1–7 was added with stirring (Scheme 1).
Point 6: A better formulation of the synthesis result analysis is needed.
Response 6: Thank you for the recommendations. We have taken into account all suggestions for improving the structuring of the material in the section Results and Discussion
Point 7: The sentence “The values of bond length and angles of the abovementioned compounds are very close to organic compounds with similar fragments.” should be confirmed by examples
Response 7: Thank you for the recommendations. The reference was placed into corresponding paragraph.
The values of bond length and angles of the abovementioned compounds are very close to organic compounds with similar fragments fragments [Allen, F.H.; Kennard, O.; Watson, D.G.; Brammer, L.; Orpen, A.G.; Taylor, R. Tables of Bond Lengths Determined by X-Ray and Neutron Diffraction. Part 1. Bond Lengths in Organic Compounds. J. Chem. Soc., Perkin Trans. 2 1987, S1–S19.]
Point 8: For IR spectra the deconvolution could give more information about how many conformers are in the mixture.
Response 8: Thank you for your question. Indeed, IR spectroscopy has a number of significant advantages over other spectral methods. In particular, in some cases it allows to determine the number of possible conformers. Исследованные нами соли растворяются только в воде. Для исключения влияния сольватационных эффектов in the present work, IR spectra were taken in a solid state with the attenuated total internal reflectance (ATR) module.
Point 9: Line 27: “subrate”
Response 9: - Technical error. Corrections have been made to the semantic part of the abstract:
Abstract: The synthesis of the products of 1,3-propanesultone ring opening during its interaction with amides of pyridinecarboxylic acids has been carried out. The dependence of the yield of the reaction products on the position (ortho-, meta-, para-) of the substituent in the heteroaromatic fragment and temperature condition was revealed. In contrast to the meta- and para-substituted substrates, the reaction involving ortho-derivatives at the boiling point of methanol unexpectedly led to the formation of a salt. On the basis of spectroscopic, X-Ray and quantum-chemical calculation data, a model of the transition-state, as well as a mechanism for this alkylation reaction of pyridine carboxamides with sultone were proposed, to explain the higher yields obtained with the nicotinamide and its N-methyl analog compared to ortho or meta parents. Based on the analysis of ESP maps, the positions of the binding sites of reagents with a potential complexing agent in space were determined. The in silico evaluation of possible biological activity showed that the synthetized compounds may revealed some promising pharmacological effects and low acute toxicity.
Point 10: Scheme 1. There is a lack of description of R1 and R2 groups.
Response 10: Thanks for the comment. Added to the description of the scheme:Scheme 1. Representative synthetic routes for compound 8–14. (The decoding of substituents R1,2 is presented in Table 1, see below).
Point 11: Table 1. It would be useful to add the synthesis details. The table description is too general.
Response 11: The task of Table 1 is to focus the reader's attention on the relationship between the structure of compounds 1-7 and the yield in the corresponding reactions. Excessive details can complicate its perception, especially since synthetic details are described in detail in Scheme 1 (reaction time added, solvents specified), as well as in the experimental part
Point 12: Figure 2. Sentences exclude each other:
Surface local minima and maxima of ESP are represented as blue and red spheres, respectively. The local surface maxima with negative ESP values are meaningless and thus not shown as”
Response 12: Figure 2. ESP mapped molecular vdW surface of nucleophilic addition reaction reagents 1–7. The unit is in kJ mol–1. Surface global minima and maxima of ESP are represented as blue and red spheres, respectively.
Point 13 and Response to Point 13 - Please see attachment
Point 14: Figure 2. The ESP charge is measured in kJ/mol?
Response 14: Our mistake. We meant ESP potential. Changes have been made

Reviewer 3 Report
The manuscript is generally interesting. The manuscript is written very well, it is clear and concise. I recommend accepting manuscript without any change.
Author Response
The authors thank the reviewer for the careful reading of the manuscript, gratefully accept all comments and suggestions for improving the manuscript
Round 2
Reviewer 2 Report
Thank you for considering some of my comments.
Line 202 fragments fragments
I do not understand russian "Исследованные нами соли растворяются только в воде. Для исключения влияния сольватационных эффектов"